# HIF-1α-Dependent Metabolic Reprogramming, Oxidative Stress, and Bioenergetic Dysfunction in SARS-CoV-2-Infected Hamsters

**DOI:** 10.3390/ijms24010558

**Published:** 2022-12-29

**Authors:** Sirsendu Jana, Michael R. Heaven, Charles B. Stauft, Tony T. Wang, Matthew C. Williams, Felice D’Agnillo, Abdu I. Alayash

**Affiliations:** 1Laboratory of Biochemistry and Vascular Biology, Office of Blood Research and Review, Center for Biologics Evaluation and Research, Food and Drug Administration, Silver Spring, MD 20993, USA; 2Laboratory of Vector Borne Viral Diseases, Office of Vaccine Research and Review, Center for Biologics Evaluation and Research, Food and Drug Administration, Silver Spring, MD 20993, USA

**Keywords:** COVID-19 disease, bioenergetics, hamster, hypoxia, proteomics

## Abstract

The mechanistic interplay between SARS-CoV-2 infection, inflammation, and oxygen homeostasis is not well defined. Here, we show that the hypoxia-inducible factor (HIF-1α) transcriptional pathway is activated, perhaps due to a lack of oxygen or an accumulation of mitochondrial reactive oxygen species (ROS) in the lungs of adult Syrian hamsters infected with SARS-CoV-2. Prominent nuclear localization of HIF-1α and increased expression of HIF-1α target proteins, including glucose transporter 1 (Glut1), lactate dehydrogenase (LDH), and pyruvate dehydrogenase kinase-1 (PDK1), were observed in areas of lung consolidation filled with infiltrating monocytes/macrophages. Upregulation of these HIF-1α target proteins was accompanied by a rise in glycolysis as measured by extracellular acidification rate (ECAR) in lung homogenates. A concomitant reduction in mitochondrial respiration was also observed as indicated by a partial loss of oxygen consumption rates (OCR) in isolated mitochondrial fractions of SARS-CoV-2-infected hamster lungs. Proteomic analysis further revealed specific deficits in the mitochondrial ATP synthase (Atp5a1) within complex V and in the ATP/ADP translocase (Slc25a4). The activation of HIF-1α in inflammatory macrophages may also drive proinflammatory cytokine production and complement activation and oxidative stress in infected lungs. Together, these findings support a role for HIF-1α as a central mediator of the metabolic reprogramming, inflammation, and bioenergetic dysfunction associated with SARS-CoV-2 infection.

## 1. Introduction

Hypoxic responses and hypoxia-mediated elements have been implicated in severe cases of COVID-19 disease, which may progress to acute respiratory distress syndrome (ARDS) and ultimately lead to end organ dysfunction and failure [1]. Under physiological conditions, both oxygen delivery by red blood cells (RBCs) and oxygen consumption by the mitochondria are highly regulated processes as part of normal oxygen homeostasis. This delicate balance is critical since either insufficient or excess oxygen leads to increased levels of damaging reactive oxygen species (ROS). The transcriptional factor HIF-1α (hypoxia-inducible factor-1α) plays a critical role in maintaining oxygen homeostasis in humans by directly controlling the expression of hundreds of target genes. Under normal oxygen tension (normoxia), the transcriptional activity of HIF-1α is halted by hydroxylation catalyzed by prolyl hydroxylase (PHD), a non-heme iron α-ketoglutarate dependent dioxygenase [2] leading to its ubiquitination and finally degradation by the proteasomal machinery. HIF contains two subunits: an α-subunit and a more stable β-subunit. During the initial stages of hypoxia, the α-subunit quickly degrades in the presence of oxygen (half-life is less than 5 min in 21% oxygen), and the β-subunit is translocated to the nucleus, where it binds to hypoxia response elements. This process results in activation of several target genes, including some that participate in glucose metabolism, control of intracellular pH, angiogenesis, erythropoiesis, and mitogenesis [2].

At the cellular level, once cells are infected with SARS-CoV-2, accumulation of HIF-1α may occur due to increased expression as well as inhibited proteasome degradation [3]. The receptor ACE-2 (angiotensin converting enzyme 2) is critical for the entry of the SARS-CoV-2 virus into cells. Under hypoxic conditions, ACE-2 is downregulated and has an inverse relationship to the concentration of HIF-1α [4]. Additionally, it has been reported that increased levels of ACE-2 were positively associated with COVID-19 infection [5]. The association between hypoxia and decreased levels of the ACE-2 viral entry receptor was proposed to be responsible for what is known as “happy hypoxia” [6].

Lack of sufficient oxygen in COVID-19 patients with ARDS is caused by alveolar-capillary barrier disruption leading to immune cell infiltration into the lungs, resulting in inhibited gas exchange [7]. Dysfunction of this critical barrier occurs in pulmonary endothelial cells treated with the S1 subunit of the spike protein without other components of the live virus [4]. In these COVID-19 patients with ARDS, the damaged lung epithelium activates self-repair processes that are insufficient to reverse the lung tissue destruction [8,9]. Severe COVID-19 lung pathology is often characterized by pneumonia and diffuse alveolar damage (DAD) with systemic inflammation accompanied by widespread vascular complications (e.g., pulmonary embolism, stroke, and blood-vessel damage) [10,11,12,13].

Since hypoxia impairs ATP production by the electron transport chain (ETC) at the terminal step of the Krebs cycle, recent inquiries have been made into alterations induced in mitochondria. The metabolic shift towards a more glycolytic metabolism mimicking “Warburg effect” appears to be involved in SARS-CoV-2 infection [14,15,16]. To date, mitochondrial dysfunction characterized by a metabolic shift to anaerobic glycolysis has been observed in peripheral blood mononuclear cells (PBMCs) from COVID-19 patients [17]. However, critical questions regarding the relationship between oxygen sensing pathways, mitochondrial metabolism, and inflammation in lungs infected with SARS-CoV-2 have remained unanswered.

Hamsters, like humans, are naturally susceptible to infection by SARS-CoV-2 since they have similar ACE-2 receptors on key target cells [18,19,20,21]. Infected hamsters normally develop respiratory disease that is moderate and resolvable, resembling the most common disease course in humans. Here, we employed an adult hamster model of COVID-19 with features that appear in humans, including the development of pneumonia, hypertrophy, and hyperplasia of bronchiolar epithelium, as well as alveolar and interstitial infiltrates [22,23]. Hamsters and humans also share similar blood oxygen affinity characteristics (P_50_ values) [24]. The findings herein support a role for HIF-1α in the metabolic reprogramming, inflammation, and mitochondrial dysfunction associated with COVID-19 disease. This is supported by the assessment of oxygen consumption rate (OCR), a measurement of mitochondrial respiration, and extracellular acidification rate (ECAR), which correlates to the number of protons released from the cell (due to contributions from glycolysis and the Krebs cycle). Furthermore, an application of an untargeted mass spectrometry-based proteomics analysis on infected hamster lung tissue revealed a deficit in ATP synthase in complex V of the ETC and metabolic reprogramming favoring glycolysis.

## 2. Results

### 2.1. Oxidative Stress and Pro-Inflammatory Cytokines in SARS-CoV-2-Infected Lungs

Syrian hamsters challenged intranasally with SARS-CoV-2 develop weight loss, respiratory distress symptoms, and lung pathology similarly to humans [23]. Consistent with previous studies, the lungs of hamsters challenged with the WA1/2020 strain show prominent staining for SARS-CoV-2 nucleocapsid protein (SNP) and severe histopathological changes characterized by multiple regions of lung consolidation with massive accumulation of infiltrates at four days post-infection (dpi) (Figure 1A–C). Areas of consolidation also exhibited damage to the alveolar epithelium as evidenced by the loss of prosurfactant C (ProSPC)-labeled Type 2 epithelial cells (Figure 1A,D). Several studies have reported heightened levels of inflammatory cytokines and oxidative stress in lung and extra-pulmonary tissues causing cellular and multi-organ damage in the pathogenesis of COVID-19 [20,21,22]. Therefore, we assayed the levels of protein carbonylation (oxidative post-translational modifications) as an indicator of oxidative stress and protein damage in the hamster lung as well as the carbonyl content in the mitochondrial fractions from the lung homogenate isolated by differential centrifugation. This indicated significantly higher protein carbonylation in SARS-CoV-2-infected lung tissue lysates compared to controls. A similar trend was observed in the mitochondrial fractions, indicating SARS-CoV-2 infection induces an oxidative stress environment in the pulmonary cells in general that also affects mitochondrial proteins (Figure 1E). Additionally, to visualize the extent of protein oxidation, proteins were derivatized with 2,4-dinitrophenylhydrazine (DNPH) and immunoblotted using anti-DNP antibody, which showed higher levels of carbonylated proteins in the virus challenged group (Figure 1F). Infected lungs also showed increased staining for 4-hydroxynonenal (4-HNE), a major end-product of lipid peroxidation, particularly in regions with epithelial damage (Appendix A). The pro-inflammatory response elicited by SARS-CoV-2 infection was further evidenced by the significant elevation of cytokine levels including IL-1β, CCL5, and CXCL10 in infected lung tissue lysates compared to uninfected controls (Figure 1G–I).

### 2.2. Nuclear HIF-1α Expression and Upregulation of Glycolytic Enzymes in SARS-CoV-2-Infected Lungs

Studies have proposed that HIF-1α regulates the hypoxic and inflammatory responses to SARS-CoV-2 infection [4,6]. Immunofluorescence analyses of infected lungs revealed extensive nuclear punctate staining for HIF-1α in cellular infiltrates localized to consolidated regions of lung tissue (Figure 2A,B). The infiltrates were identified as macrophages based on positive staining for Iba1. Minimal nuclear HIF-1α expression was observed in nonconsolidated lung regions with normal alveolar architecture, consistent with the induction of regional hypoxic and inflammatory responses in infected lungs (Figure 2A,B). Western blot analyses of nuclear extracts prepared from the hamster lung tissues identified a moderate rise in HIF-1α expression in the infected hamster lungs compared to the mock-infected controls (Figure 2C).

To examine whether the increased nuclear expression of HIF-1α correlates with the activation of HIF-1α transcriptional pathways, we analyzed the expression of key glycolytic HIF-1α target proteins including Glut1, LDH, and PDK1 by immunofluorescence and immunoblot. Immunofluorescence analyses revealed that the monocyte/macrophage population that expressed nuclear HIF-1α in regions of consolidation also expressed high levels of membrane associated Glut1, whole cell LDH, and mitochondrial-associated PDK1 (Figure 3A, Appendix A). Higher levels of Glut-1 and PDK1 were also seen in virus-infected lung tissue lysates by immunoblotting (Figure 3B). Consistent with our finding from the immunofluorescence study, we found more than a three-fold increase in specific LDH enzymatic activity in the SARS-CoV-2-infected lung tissue homogenates using an ELISA assay (Figure 3C).

### 2.3. Alteration of Mitochondrial Bioenergetics by SARS-CoV-2 Infection

Dysfunctional mitochondrial aerobic respiration is considered a risk factor for developing severe COVID-19 infection [25]. Our results on mitochondrial protein oxidation together with a significant rise in HIF-1α in the SARS-CoV-2-infected lungs (Figure 1E,F and Figure 2) also indirectly point towards existence of localized ROS generation within the mitochondrial matrix and dysfunctional aerobic metabolism. When electrons escape from the mitochondrial electron transport chain, especially at complex I or III can lead to generation of ROS within the mitochondria [26]. Therefore, we used two separate strategies to assess respiratory function in the isolated mitochondrial fractions. First, to assess the degree of coupling between the electron transport through the mitochondrial respiratory chain complexes and the terminal ATP synthesis (i.e., oxidative phosphorylation [OXPHOS]) we performed a coupling experiment using the extracellular flux analyzer (XF Assay) (Figure 4A). In a second set of experiments, we performed an electron flow assay for functional assessment of the selected mitochondrial ETC complexes together in real time (Figure 4B). The coupling assay did not reveal any major changes in basal (State 2) respiration, ADP induced (State 3) respiration, or oligomycin induced (State 4o) respiration between the two groups. However, FCC*p*-induced uncoupling led to a marginal loss of State 3u respiration in the SARS infected mitochondria (Figure 4A).

To identify any dysfunction in the mitochondrial respiratory chain components we conducted an electron flow assay. In contrast to the coupling assay, bioenergetic measurements in the electron flow assay were dependent on uncoupled mitochondria. Figure 4B shows that infection causes a reduction of the mitochondrial electron flow at different ETC complexes, indicative of an overall virus-induced impairment of mitochondrial function. We also found a lower basal OCR in the presence of pyruvate and malate, indicating compromised complex I activity in the SARS-CoV-2-infected group. This was confirmed by complete loss of oxygen consumption using rotenone as a positive control. Addition of succinate caused a rapid burst of OCR indicating electron flow through the complex II. The infected lung mitochondria showed a lower-level OCR compared to the control (Figure 4B). Addition of antimycin stopped the electron flow, further confirming the complex II activity. TMPD/ascorbate addition caused a rapid boost in OCR through complex IV. Consistent with these specific analyses of ETC complexes, we observed a similar loss in OCR in the SARS-CoV-2-infected lung tissue mitochondria compared to non-infected (Figure 4B).

Complex V, or ATP synthase, is a point of proton re-entry from the mitochondrial intermembrane space to the matrix, and inhibition of this activity could decrease mitochondrial oxidative phosphorylation capacity resulting in ROS generation [27]. In a complex V assay, we found a noticeable decrease in the ATP hydrolysis activity in the SARS-CoV-2-infected lung mitochondria (Figure 4C) indicating a possible impairment of the ATP synthesis function of complex V. To see if this mitochondrial respiration has any effect on the glycolytic metabolism, we also measured the rate of glycolytic lactate generation in real-time using an extracellular flux analyzer (XF Assay). Figure 4D shows a glycolytic rate profile, where basal glycolysis and glycolytic reserve capacity were both upregulated in the SARS-infected lungs. Together, these changes in mitochondrial bioenergetics are consistent with the observed upregulation of glycolytic enzymes, particularly PDK1. PDK1 is the gate-keeping mitochondrial enzyme that inactivates pyruvate dehydrogenase preventing the entry of pyruvate into the TCA cycle. 

### 2.4. Proteomic Characterization of Hamster Lung Tissue Lysates

To determine protein relative abundance differences in SARS-CoV-2-infected versus non-infected control hamster lung tissue, we employed a label-free micro-data independent acquisition (µDIA) strategy [28]. A database search of the resulting MS/MS tandem spectra allowed quantification of 765 proteins across the samples with 62 proteins exhibiting a statistically significant abundance change (*p*-value < 0.05). The entire protein quantification results are shown graphically (Figure 5) and as a list (Appendix A).

Suresh et al. previously described relative protein differences from infected compared to mock-infected hamsters using the isobaric tags for relative and absolute quantification (iTRAQ) methodology [18]. A limitation of this prior study was that pooled samples from infected and mock-infected hamsters were analyzed rather than biological replicates. However, a strength of the prior study was that due to using peptide sample fractionation the total number of proteins quantified were increased by two-fold compared to this study. Nevertheless, to assess the similarities and differences to the present study, we cross-referenced the list of differentially abundant proteins found in these two studies. Using the proteins with a (*p*-value < 0.05) reported by Suresh et al., our examination resulted in 127 differentially abundant proteins. Of these, 78 were quantified in our dataset with 60/78 (77%) yielding a consistent increased or decreased protein difference (Appendix A). A comparison of the 62 differentially abundant proteins found in the present study to the earlier report by Suresh and coworkers revealed that only 43 of these altered proteins were also quantified in the Suresh report. Among the 43 proteins quantified in both studies, we found that 34/43 (79%) had the same increasing or decreasing trend (Appendix A).

The complement system plays a role in the pulmonary inflammation phenotype occurring in SARS-CoV-2 patients [29], with several C3 and C5 inhibitors offering rapid clinical benefits for COVID-19 patients within three days [30]. Notably, C4a anaphylatoxin (LOC101830930) was significantly increased, and several other complement proteins (Figure 5, red highlighted proteins) were elevated but lacked statistical significance, including complement factor H (Cfh, *p*-value 0.06), complement factor I (Cfi, *p*-value 0.07), and C3/C5 convertase (Cfb). We also observed heightened levels of Cfh and Cfi, which inactivate C3b in the alternative complement pathway, in agreement with prior reports from murine models of COVID-19 infection [31]. We were unable to detect significant changes in the levels of classical complement pathway C1q receptor (Cd93) or any of the specific proteins related to the lectin pathway of complement activation, such as mannose binding lectin (Mbl) or the Mbl-associated proteases (Masp1 or Masp2). Collectively, these results indicate the alternative complement pathway is functioning in SARS-CoV-2-infected hamster lung tissue. Disease progression in severe COVID-19 patients is marked by the deposition of fibrin and collagen [7]. Notably, in the hamster model we observed upregulation of fibrinogen alpha chain (Fga), fibrinogen gamma chain isoform X2 (Fgg), and collagen alpha-1 (X) chain (Col10a1) (Figure 6A).

Increased ROS levels or their associated proteins and transcripts have been found in deceased SARS-CoV-2 patients with excessive lung neutrophil infiltration and activation [7]. Additionally, our laboratory previously observed mitochondrial ROS elevation caused by the S1 subunit of the SARS-CoV-2 spike protein [4]. Here, we found elevated levels of peroxiredoxin-1 (Prdx1), superoxide dismutase (Sod2), and glutathione S-transferase (Gstm3) (Figure 6A). These ROS-related proteomic signatures in SARS-CoV-2-infected hamsters were also supported by elevated protein carbonylation levels (Figure 1B).

Interferon-mediated immune cell activation is a hallmark pathological sign in the lung tissue of COVID-19 patients, as are the other cytokines shown in Figure 1G–I. Accordingly, a sign of interferon signaling induction was found based upon increased levels of myxovirus resistance protein 2 (Mx2) in SARS-CoV-2-infected hamsters.

Markers of hemolysis, including hemopexin (Hpx) and haptoglobin (Hp), were elevated in the SARS-CoV-2-infected hamster lung tissue in agreement with prior studies from human patients (Figure 6B) [32,33]. As a downstream result of hemolysis, we also detected a two-fold reduction in the levels of hemoglobin subunit beta (UniProt accession LOC101833678), consistent with a retrospective study of COVID-19 patients [34], although the result did not reach statistical significance (*p*-value = 0.17).

We next entered the list of 31 increased proteins (*p*-value < 0.05) in SARS-CoV-2-infected hamsters into the Kyoto Encyclopedia of Genes and Genomes (KEGG) database with the Enrichr tool (https://maayanlab.cloud/Enrichr/) [35]. A significant pathway association was observed for the complement and coagulation cascades, platelet activation, and pathways in cancer (Appendix A). Both the complement and platelet activation signaling networks were previously found by Suresh et al., however the pathways in cancer finding were not [18].

Alternatively, to identify protein signaling associated with proteins decreased by SARS-CoV-2 infection, we analyzed the list of 31 decreased proteins with the KEGG-Enrichr workflow. All the signaling networks associated with proteins downregulated were related to neurodegenerative processes (Appendix A), and none of the findings for down-regulated proteins were reported in the earlier hamster model proteome analysis [18].

### 2.5. SARS-CoV-2 Infection Reduced Levels of ATP Synthase and ADP/ATP Translocase in Hamster Lungs

Based on the impairment of the ETC documented above, we next searched our proteomic data for differentially abundant proteins related to ATP metabolism. We found significantly depleted levels of ATP synthase subunit alpha (Atp5a1) and the ADP/ATP translocase (Slc25a4) in the SARS-CoV-2-infected group (Figure 7). This ADP/ATP carrier is exported to the cytoplasm using a specialized transport protein to provide energy to the cell. Any deficiency or dysfunction in Slc25a4 or Atp5a1 leads to serious consequences for cell metabolism [36]. These critical proteins in energy production are notable, since reductions in ATP production during the human aging process has been suggested to play a role in making SARS-CoV-2 more lethal in older populations [25,37].

## 3. Discussion

COVID-19 is a complex multiorgan disease that could impair oxygen homeostatic pathways under normal oxygen tension (normoxia) and during oxygen deprivation (hypoxia). These pathways represent a delicate balance between the oxygen content of blood, oxygen sensing, and oxygen consumption in the mitochondria. The exact molecular impact of COVID-19 infection on these pathways is still not clear, however. This is particularly relevant, as these pathways (singularly or collectively) can potentially be a target for disruption by SARS-CoV-2 infection [38].

In this investigation we followed three independent experimental approaches. First, we assessed oxidative biomarkers, inflammatory responses, and mitochondrial bioenergetics in lung tissues. This was followed by tissue immunohistochemistry. Lastly, proteomic methods were used to identify common and interconnected variables in these pathways resulting from the SARS-CoV-2-infection in Syrian hamsters.

Our data collectively show that mitochondrial ROS are among the key signaling molecules that bridge the mitochondrial and oxygen sensing pathways. ROS not only create an oxidative milieu within cells, but also directly inhibit HIF-1α prolyl hydroxylation and degradation resulting in overexpression of HIF-1α in the hamster COVID-19 model pulmonary tissue analyzed. The data also document increases in the levels of a key glycolytic enzyme (lactate dehydrogenase), within the HIF-1α signaling pathway. Markers of oxidative stress triggered by the infection resulting in elevated ROS levels were matched by corresponding increases in the levels of antioxidant proteins, such as peroxiredoxin-1 (Prdx1), superoxide dismutase (Sod2), and glutathione S-transferase (Gstm3) in lung tissues. These ROS-related signatures in COVID-19 infected hamsters were also supported by elevated protein carbonylation levels. We have previously shown that the S1 subunit alone from the SARS-CoV-2 virus causes an increase in the amounts of mitochondrial ROS in cultured human pulmonary endothelial cells [4]. Until this report, we lacked an understanding on how this hyper-oxidative environment would impact cellular metabolism in situ (i.e., whole lung tissue) from the live virus. A recent report using RNA sequencing and clinical sample analyses highlighted the role of HIF-1α expression and SARS-CoV-2 ORF3a-mediated mitochondrial dysfunction in the infection and pro-inflammatory responses to COVID-19 [39].

The hamster model of COVID-19 exhibits a mild to moderate form of the disease [18,19,20,21], however, it does replicate features of lung pathology in human patients including the activation of inflammatory cytokines, elevated mitochondrial ROS, hypoxia induced HIF-1α upregulation, accumulation of thrombosis-related proteins in the blood coagulation pathway, and signs of hemolysis. In agreement with prior large-scale studies from patients and animal models of COVID-19, we found the hamster model of disease also has upregulated levels of IL-1β, CXCL5, and CXCL10 [40,41,42]. Heightened levels of IL-1β suggest there is no causal relationship between the cessation of cellular division (i.e., cellular senescence) reported in COVID-19 patient lung tissue and mitochondrial dysfunction, since mitochondrial dysfunction-associated senescence (MiDAS) involves downregulation of IL-1β. By contrast, we observed IL-1β protein levels that are 15-fold higher in COVID-19 infected hamster lungs versus mock-infected controls [7,43]. We also detected strong increases in both CXCL5 and CXCL10. CXCL5 is responsible for the recruitment of neutrophils, which are the first and most abundant leukocyte that accumulates in COVID-19 infected lung tissue. CXCL10 reported at the time of hospital admission in COVID-19 patients is the best prognostic indicator of fatality amongst 52 other possible biomarkers related to viral infection [41,42]. Finally, we observed upregulation of markers of hemolysis (haptoglobin [Hb scavenger] and hemopexin [heme scavenger]), which have been reported to be heightened in COVID-19 infected human patients [32,33].

In this investigation, we used real-time mitochondrial bioenergetic analysis by measuring OCR as a readout of oxidative phosphorylation. COVID-19 infected hamsters have an unaltered basal respiration, indicating similar mitochondrial respiratory activity equivalent to mock-infected hamsters. Protonophores like FCCP (uncoupler) can leak into the inner mitochondrial membrane and force the mitochondria to respire to their maximum capacity in an uncoupled state. In our experimental setup, uncoupled respiration (i.e., State 3u) in COVID-19 infected animals was suppressed by ~25% relative to mock-infected hamsters, indicating an exhaustion of the mitochondrial oxidative-phosphorylation capacity caused by the viral infection. It is possible this reduction in oxidative phosphorylation is more pronounced in highly inflamed lung regions since the changes in mitochondrial bioenergetics were measured using whole lung preparations and therefore reflect a combined assessment of both normal and damaged or consolidated regions of the lung.

A metabolic shift favoring glycolysis was confirmed by ECAR measurements wherein SARS-CoV-2-infected hamsters showed a 40% and 35% higher basal glycolysis and glycolytic reserve capacity, respectively, compared to mock-infected hamsters. Consistent with this proglycolytic state, we identified increased expression of key HIF-1α-regulated glycolytic proteins including Glut1, LDH, and PDK1 mainly localized to Iba1-expressing macrophages in consolidated regions of infected lungs. The highly specific and abundant nuclear expression of HIF-1α in these lesion-localized macrophages provides casual support for the transcriptional involvement of HIF-1α in these processes. These findings are consistent with previous studies that have identified HIF-1α as a key regulator of macrophage activation and metabolism, particularly in infected and inflamed tissue regions that are often hypoxic [44]. The upregulation of the membrane glucose transporter Glut1 is thought to be critical for the rapid glucose uptake that helps maintain the glycolytic activity of inflammatory-type macrophages. Activated HIF-1α also directly influences lactate metabolism by enhancing the expression of LDH, which produces lactate from pyruvate, as well as PDK1, which deactivates pyruvate dehydrogenase (PDH) by limiting the entry of pyruvate into the Krebs cycle. In agreement with this conclusion, we measured increased levels of LDH protein and enzymatic activity in the lungs of SARS-CoV-2-infected hamsters. Because mitochondrial uncoupling can drastically increase the oxidation of acetyl-CoA, thereby decreasing the acetyl-CoA concentration, it can ultimately cause activation of PDH and result in rising levels of pyruvate [45]. No significant alteration was observed for PDH or any glycolytic enzyme except for LDH. Together, these findings suggest a metabolic shift whereby HIF-1α activation reprograms cellular metabolism by altering glucose utilization from oxidative phosphorylation to glycolysis to prevent further ROS generation. This process is often referred to in oncology as the “Warburg effect,” and was reported earlier by Ajaz et al. from observations made from COVID-19 patient PBMCs [17,46].

We also identified alterations in specific respiratory complexes of the ETC that may also contribute to the observed suppression of uncoupled OCR in SARS-CoV-2-infected hamsters. The electron flow experiments revealed a drop in all different complexes (I, II and IV) in uncoupled mitochondria indicating a general loss of bioenergetic function in the virus infected group relative to controls. Further, a loss of ATP hydrolysis capacity was observed on immobilized complex V indicating a dysfunctional ATP synthase. All these findings further support the fact that high level of mitochondrial protein oxidation is most likely mediated by mitochondrial ROS. The detected ETC dysfunction prompted us to search for all mitochondrial proteins in our proteomic data with altered relative abundance in the SARS-CoV-2-infected hamsters relative to controls. This query provided more specific information on which complex V proteins may play a role in mitochondrial damage occurring in infected hamsters since the ATP synthase subunit alpha (Atp5a1) was present at only 23% of the levels in control hamsters. We also were unable to detect any of the ATP/ADP translocase (Slc25a4) that catalyzes the exchange of ADP and ATP across the mitochondrial membrane in SARS-CoV-2-infected hamsters, suggesting the levels of Slc25a4 are diminished. The ADP/ATP carrier (SLC25A4) provides energy to the cell and any deficiency or dysfunction of this membrane protein leads to serious consequences for cell metabolism [36].

In conclusion, we have shown that the HIF-1α signaling pathway is activated in the COVID-19 hamster model along with a considerable accumulation of mitochondrial ROS (Figure 8). This is characterized by a shift in cellular metabolism from oxidative phosphorylation to glycolysis. Increased HIF-1α signaling with bioenergetic switching from aerobic to anaerobic metabolism also drives the macrophage-mediated inflammatory responses in SARS-CoV-2-infected lungs (Figure 8). These observations may have implications for understanding certain features of the human disease, since mitochondrial dysfunction and its associated symptoms have been reported in patients with “Long COVID-19” [37,47]. The crosstalk between oxygen carrying, oxygen sensing (hypoxia-inducible factor) proteins, and the mitochondrial respiratory machinery could potentially provide a target for therapeutic interventions.

## 4. Methods and Materials

### 4.1. Viruses

SARS-CoV-2/human/USA/USA-WA1/2020 (GenBank: MT246667.1) was propagated in Vero E6 cells to generate working virus stocks and sequenced to confirm genotypes. Vero E6 cells (ATCC, Manassas, VA, USA) were grown in Dulbecco’s Modified Eagle’s Medium (Corning, NY, USA) supplemented with 10% fetal bovine serum (FBS) and penicillin/streptomycin.

### 4.2. Infection of Syrian Hamsters with SARS-CoV-2

All hamsters were housed and challenged by the intranasal route in BSL-3 conditions as described previously [23]. The study protocol details were approved by the White Oak Consolidated Animal Care and Use Committee and carried out in accordance with the PHS Policy on Humane Care & Use of Laboratory Animals. This study is also reported in accordance with ARRIVE Guidelines for reporting in vivo experiments [48]. Adult (aged 20 months) male Syrian hamsters (*Mesocricetus auratus*, *n* = 18) were infected by the intranasal route with 10^4^ PFU of WA1/2020 SARS-CoV-2 or PBS as a negative control (*n=* 4 per group). Specific numbers of animals used in each set of experiments are indicated in the respective figure legends. Animals were euthanized by intraperitoneal injection of 200 mg/kg Pentobarbital (Euthasol, Virbac AH, Inc., Fort Worth, TX, USA) at 4 days post infection (dpi) and lung tissues were homogenized in DMEM and heat-treated at 56 °C for 1 h prior to being taken out of BSL-3. Lung samples were also fixed in 10% neural buffered formalin for 24 h and then processed for paraffin embedding.

### 4.3. Lung Histology and Immunofluorescence Analyses

Formalin-fixed paraffin-embedded (FFPE) lung sections 4 µm thick were dewaxed, rehydrated, and stained by H&E for routine histology with pathology scores derived as previously described [49]. For immunofluorescence experiments, rehydrated FFPE lung sections were heat-treated in a microwave oven for 15 min in 10 mM sodium citrate buffer (pH 6.0) or 10 mM Tris/1 mM EDTA buffer (pH 9.0). After cooling for 30 min at room temperature, heat-retrieved sections were blocked in PBST with 2.5% bovine serum albumin (BSA) for 30 min at RT followed by overnight incubation at 4 °C with primary antibodies in 1% BSA. Primary antibodies used included SARS nucleocapsid protein (40143-MM05, Sino Biologicals, Wayne, PA, USA), prosurfactant protein C (AB3786, EMD Millipore, Burlington, MA, USA), HIF-1α ( ab51608, Abcam, Cambridge, MA, USA), Glut1 (Abcam, ab115730), LDH (Abcam, ab52488), PDK1 (ThermoFisher, MA5-32702), Iba1 (Abcam, ab5076), ATP5A (Abcam, ab14748), and 4-HNE (Abcam, ab48506). Sections were rinsed and incubated with Alexa Fluor 488 and Alexa Fluor 647-conjugated secondary antibodies for 1 h at RT (ThermoFisher, Waltham, MA, USA). Nuclei were counterstained with Hoechst 33342. For double labeling experiments, primary antibodies were mixed and incubated overnight at 4 °C. For negative controls, sections were incubated without the primary antibody or mouse and rabbit isotype antibody controls. Sections stained with conjugated secondary antibodies alone showed no specific staining. Images were captured using an Axio Observer Z1 inverted microscope (Carl Zeiss, Thornwood, NY, USA) equipped with an Axiocam 506 monochrome camera, an ApoTome.2 optical sectioning system, and a Plan-Apochromat 63×/1.4NA oil immersion with WD = 0.19 and Plan-Apochromat 20×/0.8 objective lens. Digital image post-processing and analysis were performed using the ZEN 2 ver. 2.0 imaging software. Images were constructed from Z-stack slices collected at 0.48 µm intervals (5.5 µm thickness in total) and visualized as maximum intensity projections in orthogonal mode. For semiquantitative analysis of nuclear HIF-1α expression, Z-stack images were acquired from consolidated and nonconsolidated regions for each lung section from SARS-CoV-2-infected animals (18–20 high-power fields per lung section, *n =* 4 animals). Nuclei with 20 or more clearly defined HIF-1α-stained puncta were defined as positive. Values were calculated as the percent positive HIF-1α nuclei as a function of total nuclei in each high-power field (HPF). Whole slide brightfield and fluorescence imaging was also performed using a Hamamatsu NanoZoomer 2.0-RS whole-slide digital scanner equipped with a 20× objective and a fluorescence module #L11600. Analysis software NDP.view2 was used to process images (Hamamatsu Photonics, Shizuoka, Japan).

### 4.4. Immunoblotting and Cytokine ELISA

Lung tissue lysate proteins were separated by electrophoresis using precast 4–20% NuPAGE bis-tris gels (Thermo Fisher Scientific, Waltham, MA, USA) and then transferred to PVDF membranes using iBlot system (Thermo Fisher Scientific, Waltham, MA, USA). Membranes were blocked in 5% milk-PBS and then probed with specific primary antibodies for the Western Blot analysis. Specific primary antibodies for HIF-1α, Glut-1, PDK-1 were used as indicated in the immunofluorescence analyses. Antibodies for loading controls against α-Tubulin (cat # ab7291) and Lamin B1 (ab16048) were purchased from Abcam (Cambridge, MA, USA). The pro-inflammatory cytokine CXCL10 was measured in lung homogenates using a hamster specific ELISA kit from Assaygenie (cat # HMFI0014-S, American Research Products, Inc., Waltham, MA, USA). IL-1β was measured using an ELISA kit (cat # CSB-E14259Ha) purchased from CUSABIO Technology LLC (Houston, TX, USA). Hamster CCL5 was measured by an ELISA kit (cat # MBS024469) obtained from MyBioSource, Inc. (San Diego, CA, USA). Nuclear extract was prepared by using NE-PER Nuclear and Cytoplasmic Extraction Kit (Thermo Fisher Scientific, Waltham, MA, USA). Tissue LDH activity was measured photometrically by Lactate Dehydrogenase Assay Kit (cat # ab102526, Abcam, Cambridge, MA, USA).

### 4.5. Measurement of Protein Carbonylation

Protein carbonyl content in the lung tissue lysates or in mitochondrial proteins were measured photometrically following derivatization with DNPH using a commercial kit (ab126287) from Abcam (Cambridge, MA, USA). In a similar set of experiments carbonylated proteins were also detected following immunoblotting using an anti-DNPH antibody (Abcam, ab178020).

### 4.6. Isolation of Mitochondria

Mitochondrial fraction was isolated from lung tissue homogenates by differential centrifugation. Tissue homogenate was prepared in mitochondria isolation buffer containing sucrose (70 mM), mannitol (220 mM), KH_2_PO_4_ (5 mM), MgCl_2_ (5 mM), HEPES (2 mM), EGTA (1 mM) and BSA (0.2%) and all subsequent steps of the preparation were performed on ice. Homogenate was first centrifuged at 1000× *g* for 10 min at 4 °C. The supernatant was collected and centrifuged at 12,000× *g* for 10 min at 4 °C to obtain mitochondrial pellet. The pellet was resuspended in the same buffer and centrifuged again at 12,000× *g* for 10 min at 4 °C. The pellet was then washed with a BSA-free mitochondria isolation buffer. Finally, the pellet was resuspended in BSA-free mitochondria isolation buffer and aliquoted in the desired volume. Total protein (mg/mL) was determined using Pierce BCA Protein Assay Reagent (Thermo Fisher Scientific, Waltham, MA, USA). Mitochondrial preparations were used immediately to measure oxygen consumption rate (OCR) or kept at −80 °C for further mitochondrial complex assays.

### 4.7. Mitochondrial Bioenergetic Measurements

Bioenergetic function in isolated mitochondria was measured using an Agilent-Seahorse XF24 Extracellular Flux Analyzer (Seahorse Biosciences, North Billerica, MA, USA). Briefly, isolated mitochondria were plated on a 24-well Seahorse V7 plate, and the plate was centrifuged at 2500× *g* at 4 °C using a swing rotor to adhere the mitochondria to the bottom of the wells. The XF assay was performed at 37 °C following manufacturer’s protocol for isolated mitochondria (Agilent Seahorse Biosciences, North Billerica, MA, USA). After an initial recording of basal oxygen consumption rate (OCR) from mitochondria without substrate, a mixture of succinate (5 mM) and ADP (1 mM) was added directly through an automated injection to initiate substrate induced state 3 respiration/OCR. Mitochondrial basal and state 3 respiration were calculated from the OCR plot generated by the XF 24 analyzer.

In a Coupling Assay, mitochondrial respiration (10 μg/well) was sequentially measured in a coupled state with substrate present (basal respiration, State 2), followed by State 3 (phosphorylating respiration, in the presence of adenosine 5′-diphosphate sodium salt [ADP] and substrate), State 4 (non-phosphorylating/resting respiration) following conversion of ADP to adenosine triphosphate (ATP), State 4o, induced with the addition of oligomycin. Next, maximal uncoupler-stimulated respiration (State 3u) was detected by the administration of the uncoupling agent carbonylcyanide-4-trifluorometh-oxyphenyl hydrazone (FCCP). At the end of the experiment the Complex III inhibitor, antimycin A, was applied to completely shut down the mitochondrial respiration. This ‘coupling assay’ examines the degree of coupling between the ETC and the OXPHOS and can distinguish between ETC and OXPHOS with respect to mitochondrial function/dysfunction. In a second set of studies electron flow experiments were conducted. This method allows the functional assessment of selected mitochondrial complexes together in the same time frame. Mitochondrial electron transport was stimulated by the addition of pyruvate/malate (10 mM/2 mM, respectively, in order to enable the activity of all complexes), with succinate (10 mM, in the presence of the Complex I inhibitor rotenone, 2 μM, in order to direct the electron flow exclusively through complexes II, III and IV), or with the artificial substrates ascorbate/TMPD (10 mM/100 μM, respectively, in the presence of the Complex III inhibitor antimycin at 4 μM, in order to selectively activate Complex IV).

### 4.8. Mitochondrial ATP Synthase (Complex V) Activity

An aliquot of the mitochondrial suspension (20 μg of protein) was utilized to measure mitochondrial ATP Synthase (Complex V) activity. Quantitative measurement of the activity of Complex V was measured photometrically using an ATP synthase Enzyme Activity Microplate Assay Kit (ab109714, Abcam, Cambridge, MA, USA). The Complex V was immunocaptured within the microplate wells. The enzymatic activity of ATP synthase converting ATP to ADP was then measured by a coupled oxidation reaction of NADH to NAD+ with a reduction in absorbance at 340 nm using a SYNERGY/HTX multimode reader, BioTek Instruments (Winooski, VT, USA). Relative activity was expressed as changes in absorbance per minute per μg protein.

### 4.9. µDIA Mass Spectrometry Sample Preparation and Acquisition

Whole hamster lung tissues were placed in sodium dodecyl sulfide (SDS) to a final concentration of 1 % (*v*/*v*). The samples were homogenized by pipetting each sample through ten cycles of resuspension with a 200 µL pipette until the tissues were solubilized. Each sample was then boiled for 5 min at 99 °C. An amount corresponding to 50 µg of total protein estimated by the BCA assay from each sample was subjected to SDS-PAGE electrophoresis for 5 min followed by an in-gel trypsin digestion with iodoacetamide alkylation as described before [50]. Peptide digestions from each sample were analyzed by LC-MS/MS on a Q-Exactive Orbitrap mass spectrometer (Thermo Scientific, Waltham, MA, USA) in conjunction with an EASY nLC 1200 HPLC (Thermo Scientific, Waltham, MA, USA) and an EASY-Spray nano-electrospray ionization source operating in positive ion mode. Peptides were separated using a 75 µm × 250 mm 2 µm C18 reverse phase analytical column (Thermo Scientific, Waltham, MA, USA). Peptide elution was performed with an increasing percentage of acetonitrile over a 120 min gradient with a flow rate of 300 mL/min. The mass spectrometer was operated in data-independent acquisition (DIA) mode with MS2 spectra acquired at 34 distinct 16 *m*/*z* shifted mass windows stepping from 400–940 *m*/*z* with an MS survey scan obtained once every duty cycle from 400–950 *m*/*z*. MS and MS2 spectra were acquired with an Orbitrap scan resolution of 140,000 and 15,000, respectively. The accumulation gain control (AGC) was set to 1 × 10^6^ and 1 × 10^5^ ions with a maximum injection time of 100 and 80 milliseconds for MS and MS2 scans, respectively. The precursor ions in MS2 fragment ion scans were selected across a mass isolation window of 24 *m*/*z* and dissociated by HCD (High Energy Collisional Dissociation) with a 30% normalized collision energy.

### 4.10. µDIA Proteomic Data Analysis

Raw mass spectra were analyzed using Protalizer µDIA software (v1.1.3.2) from Vulcan Biosciences (Birmingham, AL, USA) [28]. Peptide and protein identifications were made using the Protein Farmer search engine against the reviewed and unreviewed forward and reversed *Mesocricetus auratus* UniProt database (2021—10 release). Mass spectra were identified and quantified with a 12 ppm fragment ion mass tolerance after mass calibration. Carbamidomethylation of cysteine residues was searched as a fixed modification in all analyses. Peptides with one trypsin missed cleavage were included in the analysis. A strict false discovery rate (FDR) based on a reversed database search of 1% at the peptide level and 5% at the protein level was applied for each sample analyzed. Normalized peptide and protein quantitative relative abundance values were calculated by normalizing the MS2 chromatogram sum intensity of each peptide to the total sum intensity of all peptide MS2 chromatograms in each sample followed by Log2 transforming the values and applying a two-tailed unpaired *t*-test as described previously [28].

### 4.11. Statistical Analysis

All statistical calculations and data plotting were done using GraphPad Prism, version7 software. All values are expressed as mean ± SEM. Statistical calculations were made using either Student’s *t*-test or Mann–Whitney test, two-tailed, where *p* < 0.05 was considered significant between groups.

## Figures and Tables

**Figure 1 ijms-24-00558-f001:**
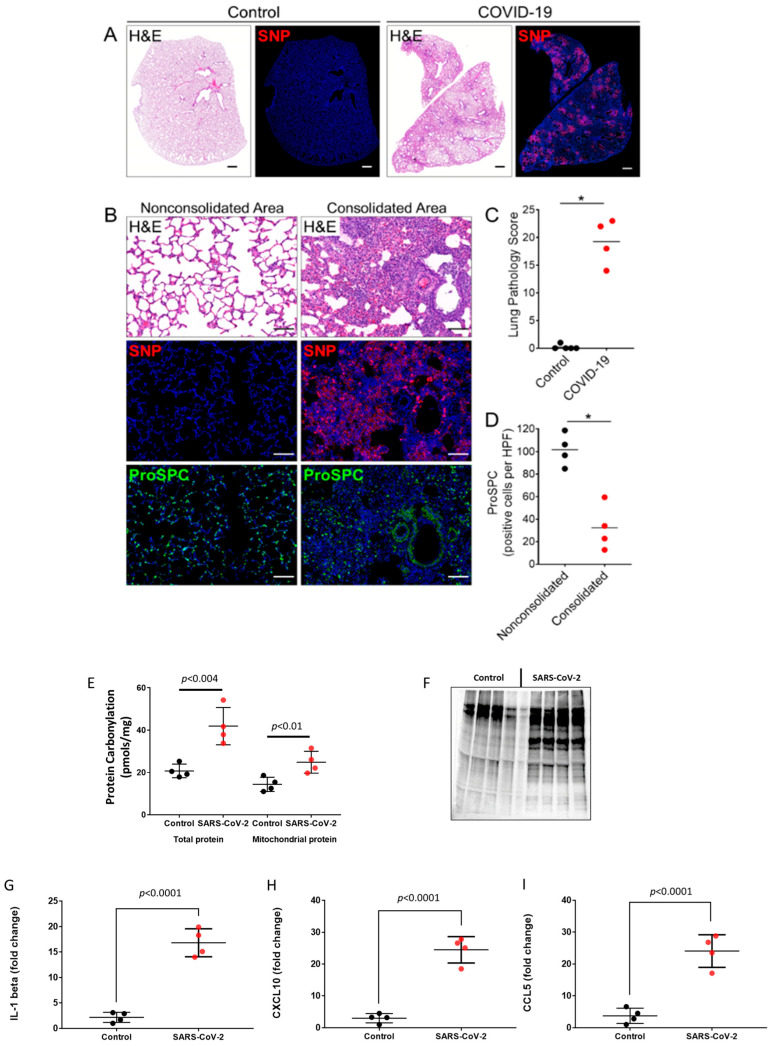
SARS-CoV-2 infection and lung pathology in Syrian hamsters. (**A**) Representative staining for H&E and SARS-CoV-2 nucleocapsid protein (SNP) in serial lung sections from an uninfected hamster (control) and a SARS-CoV-2-infected hamster at 4 days post-infection (dpi). Scale bars: 1 mm. (**B**) Staining for H&E, SNP, and prosurfactant protein C (ProSPC) in consolidated and nonconsolidated regions of infected lungs at 4 dpi. The consolidated area shows extensive accumulation of cellular infiltrates and epithelial debris with marked SNP staining and a reduced number of ProSPC-expressing alveolar type 2 cells (AT2). The nonconsolidated region shows normal alveolar architecture, absence of SNP staining, and a normal distribution of ProSPC-positive AT2 cells. Nuclei were counterstained with Hoechst 33342 dye (blue). Scale bars: 100 µm. (**C**) Histopathological scoring of lungs from uninfected (*n =* 5) and SARS-CoV-2 hamsters 4 dpi (*n =* 4). Values are means ± SEM. * *p <* 0.05. (**D**) Semiquantitative analysis of ProSPC-expressing cells in consolidated and non-consolidated regions of infected lungs at 4 dpi. ProSPC expressing cells were counted in 10–15 consolidated or nonconsolidated fields per lung for each hamster (*n =* 4). Values were averaged and shown as the means per high power field ± SEM for each group. (**E**) Bar graphs showing levels of protein carbonylation in total lung homogenate and in isolated mitochondrial fractions. (**F**) Levels of protein carbonylation in total lung homogenates were also detected by immunoblotting using anti-DNP antibody as described in the methods section. (**G**–**I**) Levels of pro-inflammatory cytokines in lung tissue lysates were detected by ELISA, *p* < 0.05 vs. corresponding controls; Values are means ± SD.

**Figure 2 ijms-24-00558-f002:**
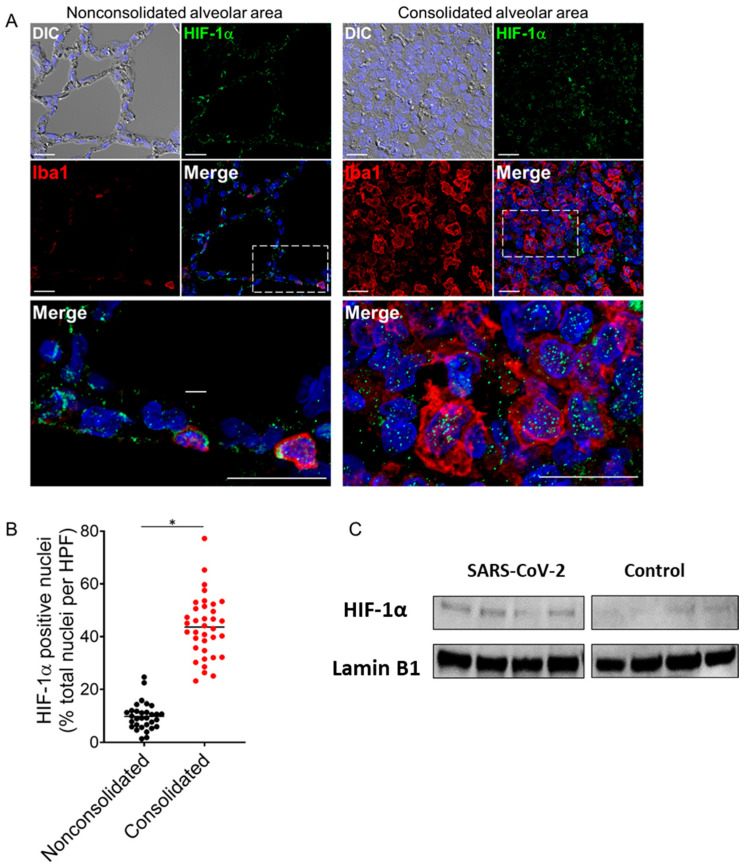
Activation of HIF-1α in lung monocytes/macrophages of infected hamsters. (**A**) Double immunofluorescence imaging of HIF-1α and Iba1 (monocyte/macrophage marker) in lung sections from a SARS-CoV-2-infected hamster at 4 dpi. Prominent HIF-1α staining localizes to the nuclei of Iba1-positive monocytes/macrophages that accumulate in areas of lung consolidation. Minimal nuclear HIF-1α expression is observed in nonconsolidated lung regions with normal alveolar architecture. Nuclei were counterstained with Hoechst 33342 (blue). Scale bars: 20 µm. (**B**) Semiquantitative analysis of HIF-1α expression in consolidated and nonconsolidated regions of SARS-CoV-2-infected lungs at 4 dpi. A total of 8–10 high power fields (HPFs) from consolidated and nonconsolidated regions were analyzed per lung section for each infected animal (*n =* 4). Plotted values represent the percent positive HIF-1α nuclei as a function of total nuclei for each HPF. * *p* < 0.05, Mann–Whitney test. (**C**) Immunoblotting showing HIF-1α expression in nuclear extracts from lung tissue lysates of SARS-CoV-2-infected and non-infected hamsters (upper panel) and Lamin B1 as loading control (lower panel). Densitometric analysis for relative expression of HIF-1α was done by BioRad Image Lab 5.2.1 software (Appendix A). DIC = differential interference contrast.

**Figure 3 ijms-24-00558-f003:**
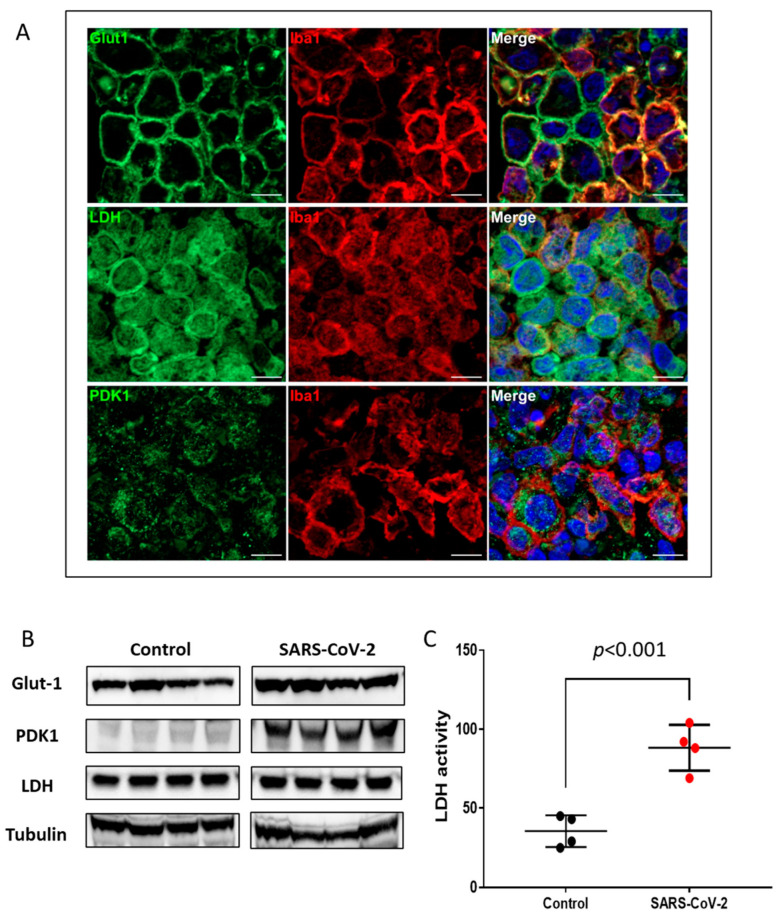
Upregulation of HIF-1α-regulated glycolytic enzymes. (**A**) Double immunofluorescence imaging of Iba1 combined with Glut1, LDH, or PDK1 in lung sections from a SARS-CoV-2-infected hamster at 4 dpi. Scale bars: 10 µm. (**B**) Representative Western blot images of Glut-1, PDK1, and LDH in total lung homogenates of non-infected (control) and SARS-CoV-2-infected hamsters (*n* = 4). Equal protein loading was confirmed by reprobing the blots with Tubulin (lower panel). Densitometric analyses for relative expression of Glut-1, PDK1, and LDH were done by BioRad Image Lab 5.2.1 software (Appendix A). (**C**) LDH enzymatic activity in uninfected (control) and SARS-CoV-2-infected lung tissue lysates was measured by ELISA assay. Values are mean ± SD. *n* = 4.

**Figure 4 ijms-24-00558-f004:**
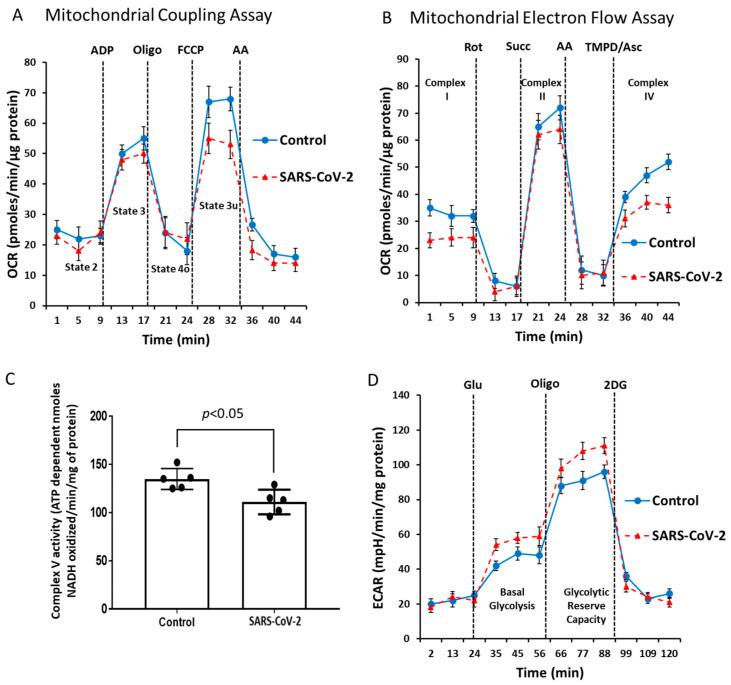
SARS-CoV-2 infection causes impairment of mitochondrial bioenergetics. Mitochondrial respiration/function was determined by oxygen consumption rate (OCR) measured by the Seahorse XF24 Extracellular Flux Analyzer (Seahorse Bioscience). (**A**) During the coupling experiment with isolated mitochondria, the assay media contains succinate as a complex II substrate and rotenone as a complex I inhibitor. Sequential measurement of basal, State 2, State 3, State 4o, and the uncoupler-stimulated respiration, State 3u, were performed through the sequential injections of ADP, oligomycin, FCCP, and antimycin. (**B**) An Electron-Flow assay demonstrates the electron flow activity through different complexes of the ETC. The assay media contains pyruvate/malate as substrates of complex I, and FCCP to uncouple the mitochondrial function. In uncoupled mitochondria all complexes were examined individually by the sequential injection of rotenone, succinate, antimycin, and TMPD/ascorbate to monitor the inhibition of complex I, activity of complex II, inhibition of complex II, and activity of complex IV, respectively. (**C**) Bar graph showing mitochondrial ATP synthase (complex V) activity. (**D**) A glycolytic activity profile was monitored by measuring ECAR using a Seahorse XF24 Extracellular Flux Analyzer. Basal glycolysis and glycolytic reserve capacity were measured by sequential addition of glucose, oligomycin, and 2-DG in lung tissue lysates. Values are means ± SD.

**Figure 5 ijms-24-00558-f005:**
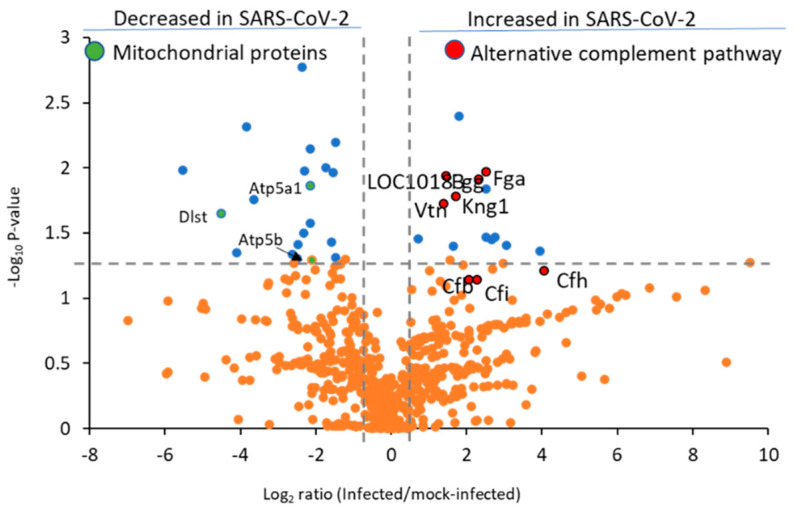
Complement and coagulation cascade proteins are among the most upregulated by the SARS-CoV-2 virus in hamster lung tissue. Data shown are untargeted quantification results from 607 proteins (158 proteins are listed in Appendix A but were detected in only infected or control hamsters and are not included in this plot). The blue dots represent proteins with *p* < 0.05. and orange dots show proteins with *p* > 0.05. Complement and coagulation pathway proteins are highlighted in red and mitochondrial proteins in green with UniProt gene abbreviations. The vertical lines denote fold-change values of −1.5 and 1.5 (without Log2 transformation), while the horizonal line shows a *p*-value of 0.05 (without −Log10 transformation).

**Figure 6 ijms-24-00558-f006:**
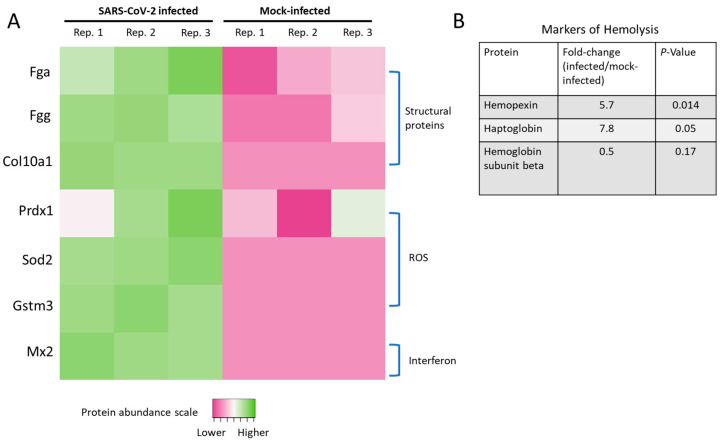
Structural proteins, ROS, and altered interferon-signaling in infected hamsters. (**A**) Heatmap of proteins documented to have differences in SARS-CoV-2-infected patients or models. Data shown are the Log_2_ protein relative abundance values per biological replicate. Proteins below the limit of quantification in any particular sample were assigned −1 (dark pink in the heatmap, which included all of the mock-infected replicates for Sod2, Gstm3, and Mx2). (**B**) Levels of hemolysis markers changed in SARS-CoV-2-infected hamsters.

**Figure 7 ijms-24-00558-f007:**
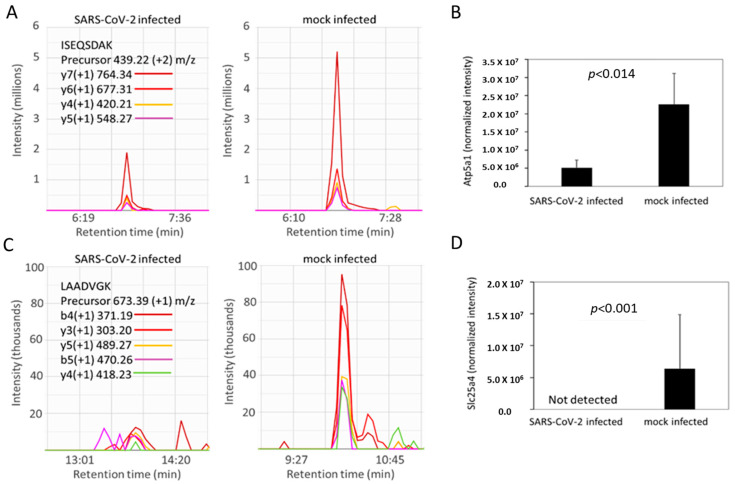
Atp5a1 (ATP synthase subunit alpha) and Slc25a4 (ADP/ATP translocase) are decreased in SARS-CoV-2-infected hamsters. (**A**,**C**) Representative MS2 spectra of infected versus control hamsters for a peptide assigned to Atp5a1 (panel (**A**), tryptic peptide sequence ISEQSDAK corresponding to residues 532–539 in UniProt accession number F1T2M2) and Slc25a4 (panel **C**, tryptic peptide sequence LAADVGK from residues 141–147 in UniProt accession number A0A1U7QFU8) in infected (left) versus control (right) hamsters. (**B**,**D**) Bar graphs show mean values for the mass spectrometry-based quantification of Atp5a1 from 21 peptides (panel (**B**) and Slc25a4 from 3 peptides panel (**D**)). Statistical analysis was by a Student’s *t* test as described in the Methods. Error bars indicate SD.

**Figure 8 ijms-24-00558-f008:**
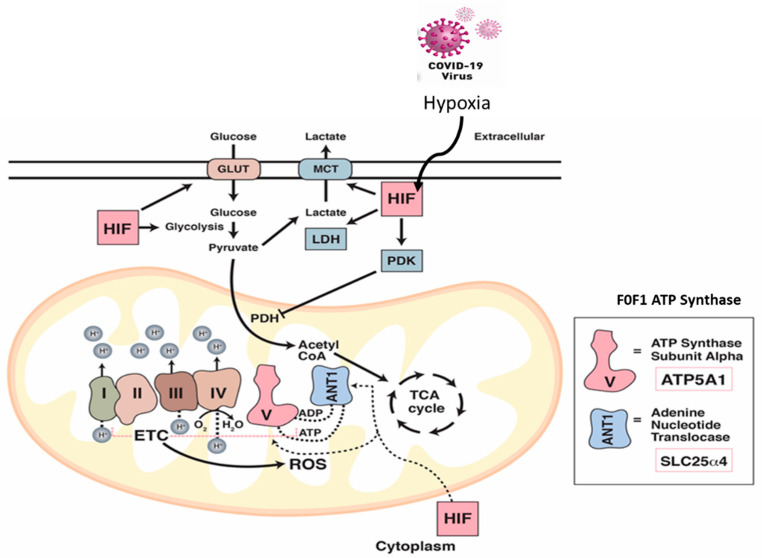
The impact of COVID-19 infection on oxygen homeostasis. Under normal aerobic conditions, HIF-1 regulates glucose metabolism resulting in pyruvate formation. Pyruvate dehydrogenase (PDH) converts pyruvate to acetyl-CoA for the entry into the TCA cycle. Under hypoxic and/or COVID-19 infection conditions, HIF-1 targets and activates many glycolytic enzymes including lactate dehydrogenase (LDH) to increase lactate production. HIF-1 also promotes pyruvate dehydrogenase kinase (PDK) to inactivate PDH which prohibits the conversion of pyruvate to acetyl-CoA. Under these infection conditions, the ETC becomes a source for reactive oxygen species (ROS) which stabilize HIF-1. Impairment of the ETC leads to significant depletion in the levels of ATP synthase subunit alpha (Atp5a1) responsible for producing ATP through mitochondrial complex V, and the ADP/ATP translocase (Slc25a4). Both proteins are regulated via HIF-1α.

## Data Availability

The mass spectrometry proteomics data have been deposited to the Proteo-meXchange Consortium via the PRIDE partner repository with the dataset identifier PXD036862 [51]. All other datasets used and/or analyzed during the current study are available from the corresponding author on reasonable request.

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
