# Peer review of "HIF-1α-Dependent Metabolic Reprogramming, Oxidative Stress, and Bioenergetic Dysfunction in SARS-CoV-2-Infected Hamsters"

_ijms, 2022, doi:10.3390/ijms24010558_

Round 1
Reviewer 1 Report
Manuscript review
HIF-1α-dependent metabolic reprogramming, oxidative stress, and bioenergetic dysfunction in the lungs of hamsters with COVID-19
Sirsendu Jana, Michael R. Heaven, Charles B. Stauft, Tony T. Wang, Matthew C. Williams, Felice D’Agnillo and Abdu I. Alayash
In their manuscript "HIF-1α-dependent metabolic reprogramming, oxidative stress, and bioenergetic dysfunction in the lungs of hamsters with COVID-19 ", Sirsendu Jana et al. are trying to determine the role that HIF-1α plays in different metabolic disorders associated with COVID-19 disease. Several aspects are studied, the production of ROS, variations in the expression of some glycolytic enzymes, cytokines, alterations in mitochondrial bioenergetics and finally a proteomic analysis on lung tissues infected or not by SARS-CoV-2. The authors propose that in the hamster model of COVID-19, the HIF-1α signaling pathway is activated along with a significant accumulation of mitochondrial ROS, a phenomenon leading to a shift in cellular metabolism from oxidative phosphorylation to glycolysis.
The manuscript is well organized, the data presented are clear and well done, and the conclusions appear to be largely consistent with the results. However, proteomic experiments described in the manuscript appear to be partially overlapping with the published literature, which reduces the value of the study. Moreover, there are several points that could be improved (see below).
Major points:
- The authors propose an increased in ROS production in lung tissues infected with SARS-CoV-2. Several separate results seem to support this ROS production, but always by indirect measurements. A direct ROS measurement would have been much more informative, and this is relatively easy to perform.
- The authors present Figure 4C a small decrease in ATP synthase activity of about 20% that is not related to the 75% decrease in ATP5a1 subunit that they observe by the proteomic approach. The authors do not comment about the discrepancy.
- In the same vein, Figure 4C, the “y” axis legend lacks unit and very little information were provided in the materials and methods about the technique used. The system used is based on measuring the hydrolysis activity of ATP synthase, but the conclusions of the manuscript refer to the synthesis activity. There are many examples in the literature where hydrolysis activity is not altered while synthesis is and vice versa. It is likely that the conclusions are correct, and that the measurement of hydrolysis does not accurately reflect the low level of ATP5a1 expression.
- The manuscript does not contain the supplemental Table 1, 2 and 3, only Table 4 is presented. In order to study the paper properly, it is essential to have this information.
- In Figure 7, the quantification of ATP5a1 subunit and Slc25a4 is quite difficult to understand, with a legend that is not very helpful. It also seems to be based on only one peptide. Slc25a4 also presents a very wide error bar, which limits the interpretation of this data.
Minor points:
- Line 170: "punctate cytosolic expression of mitochondrial enzyme PDK1". It is very surprising to have a cytosolic localization for a mitochondrial protein; we rather expect to see the mitochondrial network labeled. To confirm this localization, it would be interesting to use another mitochondrial marker.
- Line 190: No reference for this statement: "Inhibition of any complexes of the mitochondrial respiratory chain can lead to generation of ROS within the mitochondria". I think the examples are mostly related to complex I and III.
Author Response
Reviewer 1:
In their manuscript "HIF-1α-dependent metabolic reprogramming, oxidative stress, and bioenergetic dysfunction in the lungs of hamsters with COVID-19 ", Sirsendu Jana et al. are trying to determine the role that HIF-1α plays in different metabolic disorders associated with COVID-19 disease. Several aspects are studied, the production of ROS, variations in the expression of some glycolytic enzymes, cytokines, alterations in mitochondrial bioenergetics and finally a proteomic analysis on lung tissues infected or not by SARS-CoV-2. The authors propose that in the hamster model of COVID-19, the HIF-1α signaling pathway is activated along with a significant accumulation of mitochondrial ROS, a phenomenon leading to a shift in cellular metabolism from oxidative phosphorylation to glycolysis.
The manuscript is well organized, the data presented are clear and well done, and the conclusions appear to be largely consistent with the results. However, proteomic experiments described in the manuscript appear to be partially overlapping with the published literature, which reduces the value of the study. Moreover, there are several points that could be improved (see below).
Response:
In the current text we described the differences between the iTRAQ methodology used by Suresh et al and our label-free micro-data independent acquisition (µDIA) strategy (Lines 255-270). The results of the proteomic study do largely agree with a reference cited in our manuscript by Suresh et al. But our work has confirmed and extended these prior findings, which are made clear to readers in Supplemental Tables 2-3. An important element that strengthens the claims in our study is that our results were generated from biological replicates as opposed to technical replicates of pooled samples as reported in the Suresh et al. study.
Major points:
- The authors propose an increased in ROS production in lung tissues infected with SARS-CoV-2. Several separate results seem to support this ROS production, but always by indirect measurements. A direct ROS measurement would have been much more informative, and this is relatively easy to perform.
Response:
To address this reviewer’s comment, we carried out new experiments in which lung sections were immunostained for the oxidative marker 4-hydroxynonenal (4-HNE), a major end product of lipid peroxidation. Infected lungs showed increased 4-HNE staining compared to uninfected control lungs consistent with the other evidence of ROS production in the manuscript (Supplementary Fig. 1).
In the Results. Line 113, we added: “Infected lungs also showed increased staining for 4-hydroxynonenal (4-HNE), a major end-product of lipid peroxidation, particularly in regions with epithelial damage (Supplementary Fig. 1).”
- The authors present Figure 4C a small decrease in ATP synthase activity of about 20% that is not related to the 75% decrease in ATP5a1 subunit that they observe by the proteomic approach. The authors do not comment about the discrepancy.
Response:
We accept the discrepancy, but we don’t think the direct comparison between peptide abundance level of a particular protein observed in mass-spectrometric analysis and the actual biochemical / enzymatic activity of a protein is justifiable. The same is true for mRNA expression of a protein vs. its functional activity. Actual function of a protein may depend upon several factors like post-translational modifications, sub-cellular localization, and protein-protein interactions etc. We think, even with a significant loss of functional ATP synthase protein molecules, a smaller percentage of active molecules can maintain an optimum activity.
- In the same vein, Figure 4C, the “y” axis legend lacks unit and very little information were provided in the materials and methods about the technique used. The system used is based on measuring the hydrolysis activity of ATP synthase, but the conclusions of the manuscript refer to the synthesis activity. There are many examples in the literature where hydrolysis activity is not altered while synthesis is and vice versa. It is likely that the conclusions are correct, and that the measurement of hydrolysis does not accurately reflect the low level of ATP5a1 expression.
Response:
We are thankful to the reviewer for correcting on this point. We have corrected the figure with proper Y-axis legend and elaborated the experimental method.
Complex V activity assay that we used, measures the activity of complex V as an ATPase (hydrolysis), since the ATP synthase (production) reaction typically requires freshly isolated, coupled mitochondria. Since, within the intact mitochondria, this same enzyme generates ATP due to the presence of proton gradient, we assume that any defect in the ATP synthase enzyme will also affect the ATP generation capacity. This is the most common approach adopted by many researchers. However, we thank the reviewer and we have corrected both the Results and Discussion sections to specifically indicate that ATP hydrolysis activity in the SARS-CoV2 infected group is impacted.
We have added: ‘In a complex V assay, we found a noticeable decrease in the ATP hydrolysis activity in the SARS-CoV-2 infected lung mitochondria (Fig. 4C) indicating a possible impairment of the ATP synthesis function of complex V’.
In the Discussion section, we added ‘Further, a loss of ATP hydrolysis capacity was observed on immobilized complex V indi-cating a dysfunctional ATP synthase.’
- The manuscript does not contain the supplemental Table 1, 2 and 3, only Table 4 is presented. In order to study the paper properly, it is essential to have this information.
Response:
We are sorry for the missing supplementary tables. We encountered a technical problem for uploading those large Excel files. We informed the editorial office, and we will make sure to upload the files properly with this revised version.
- In Figure 7, the quantification of ATP5a1 subunit and Slc25a4 is quite difficult to understand, with a legend that is not very helpful. It also seems to be based on only one peptide. Slc25a4 also presents a very wide error bar, which limits the interpretation of this data.
Response:
We have revised Figure 7 legend in our manuscript to clarify the peptides shown. The purpose of Figure 7 was to highlight two mitochondrial proteins downregulated in our proteomic dataset that may be related to the bioenergetic changes observed in the manuscript. Only one representative peptide example for each protein was provided as a summary of the data for the reader. The ATP5a1 quantification was based on 21 peptides unique to that protein entry in the UniProt database used for the search as shown in Supplemental Table 1. While Slc25a4 quantification was based on 3 peptides as presented in Supplemental Table 1. Additionally, to make our proteomic data as transparent as possible to readers of the article all the raw and processed files are publicly available in the ProteomeXchange consortium with the dataset identifier PXD036862.
Minor points:
- Line 170: "punctate cytosolic expression of mitochondrial enzyme PDK1". It is very surprising to have a cytosolic localization for a mitochondrial protein; we rather expect to see the mitochondrial network labeled. To confirm this localization, it would be interesting to use another mitochondrial marker.
Response:
To address the reviewer’s comment, we performed additional dual-staining experiments for PDK1 and the mitochondrial membrane enzyme, ATP5A. These experiments confirm the high-level localization of PDK1 with the mitochondrial network (Supplementary Fig. 2). We have also clarified this passage in the manuscript text.
In the Results, the sentence “.. and exhibited punctate cytosolic expression of the mitochondrial enzyme PDK1 (Fig. 3A)” was replaced with “ …and mitochondrial-associated PDK1 (Fig. 3A, Supplementary Fig. 2)”
- Line 190: No reference for this statement: "Inhibition of any complexes of the mitochondrial respiratory chain can lead to generation of ROS within the mitochondria". I think the examples are mostly related to complex I and III.
Response:
We agree with the reviewer, and we have clarified this passage in the manuscript text. We have inserted the following sentence with a reference,
‘When electrons escape from the mitochondrial electron transport chain, especially at complexes I or III can lead to generation of ROS within the mitochondria (Ref: Pelicano, JBC, 2003)’
Reviewer 2 Report
Paper is interesting and having good findings. Hwever authors needs to fix these issues.
1. Please shorten the title: HIF-1α-dependent metabolic reprogramming, oxidative stress, 2 and bioenergetic dysfunction in the lungs of hamsters with COVID-19 and it looks very heavy with the given concept
2. Rephrase these lines in abstract-A concomi- 19 tant marginal reduction in mitochondrial respiration was also observed as seen by a partial loss of oxygen consumption rates (OCR) in both coupled and uncoupled states of respiration in isolated 21 mitochondrial fractions of SARS-CoV-2 infected hamster lungs. Proteomics analysis revealed specific deficits in the mitochondrial ATP synthase (Atp5a1) within complex V and in the ATP/ADP translocase (Slc25a4). The statements do not have smooth flow
3. Incorporate data findings in the abstract before closure
4. avoid these type of sentence constructions-This is followed by ubiquitination and degradation by the proteasome. Lack of sufficient oxygen in COVID-19 patients with ARDS is caused by alveolar capillary barrier disruption leading to immune cell infiltration into the lungs, resulting in inhibited gas exchange
5. 2.1. SARS-CoV-2 induces oxidative stress and inflammatory cytokines needs robustic explanation for its findings
6. Figure 2C, replace Lamin B1 blot with linear fashioned blot. Same is true with Figure 3B i.e Tubulin
7. Figure 3B requires better legend than stereotyped presentation
8. Figure 7 requires good legend by highlighting its findings.
Author Response
Reviewer 2:
Comments and Suggestions for Authors
Paper is interesting and having good findings. Hwever authors needs to fix these issues.
- Please shorten the title: HIF-1α-dependent metabolic reprogramming, oxidative stress, and bioenergetic dysfunction in the lungs of hamsters with COVID-19 and it looks very heavy with the given concept
Response:
We have shortened the title by removing “in the lungs”. The new title reads “HIF-1α-dependent metabolic reprogramming, oxidative stress, and bioenergetic dysfunction in SARS-CoV-2 infected hamsters”.
- Rephrase these lines in abstract-A concomitant marginal reduction in mitochondrial respiration was also observed as seen by a partial loss of oxygen consumption rates (OCR) in both coupled and uncoupled states of respiration in isolated mitochondrial fractions of SARS-CoV-2 infected hamster lungs. Proteomics analysis revealed specific deficits in the mitochondrial ATP synthase (Atp5a1) within complex V and in the ATP/ADP translocase (Slc25a4). The statements do not have smooth flow
Response:
We have shortened the sentences as ‘A concomitant reduction in mitochondrial respiration was also observed as indicated by a partial loss of oxygen consumption rates (OCR) in isolated mitochondrial fractions of SARS-CoV-2 infected hamster lungs. Proteomic analysis further revealed specific deficits in the mitochondrial ATP synthase (Atp5a1) within complex V and in the ATP/ADP translocase (Slc25a4).’
- Incorporate data findings in the abstract before closure
Response:
The abstract in its current form describes fully the mechanistic interplay between the key elements of the study; SARS-CoV-2 infection, inflammation, and oxygen homeostasis (such as oxygen sensing and respiratory mitochondrial mechanisms). We believe the data is sufficiently described prior to end of the abstract.
- avoid these type of sentence constructions-This is followed by ubiquitination and degradation by the proteasome. Lack of sufficient oxygen in COVID-19 patients with ARDS is caused by alveolar capillary barrier disruption leading to immune cell infiltration into the lungs, resulting in inhibited gas exchange
Response:
We have now rephrased the sentence as ‘….leading to its ubiquitination and finally degradation by the proteasomal machinery.’
- 2.1. SARS-CoV-2 induces oxidative stress and inflammatory cytokines needs robustic explanation for its findings
Response:
We do not understand what the reviewer means by ‘robustic explanation’ as the data described in this section fully supports and is consistent with the subheading. However, possibly the reviewer is asking us to clarify or rephrase the subheading. We changed the section subheading from ‘SARS-CoV-2 induces oxidative stress and inflammatory cytokines’ to ‘Oxidative stress and pro-inflammatory cytokines in SARS-CoV-2 infected lungs.’
- Figure 2C, replace Lamin B1 blot with linear fashioned blot. Same is true with Figure 3B i.e Tubulin
Response:
We acknowledge that the Lamin B1 and Tubulin blots are somehow distorted. We thank the reviewer for catching this. We have changed the orientation of the blots for a better representation.
- Figure 3B requires better legend than stereotyped presentation
Response:
We thank the reviewer for this suggestion. We have made the figure 3B legend more descriptive.
- Figure 7 requires good legend by highlighting its findings.
Response:
The Figure 7 legend has been revised for clarity to the reader in our resubmitted manuscript. The purpose of Figure 7 was to show a relative difference in two mitochondrial proteins detected by the mass spectrometry approach used. It shows both mitochondrial proteins are depleted, and further details are provided in Supplemental Table 1. We have also strived to make our data as transparent as possible to readers by depositing all the mass spectrometry results and raw files in the publicly accessible ProteomeXchange with the dataset identifier PXD036862 listed in our manuscript.
Round 2
Reviewer 1 Report
No further comments. The authors have answered the questions.